# Genomic Epidemiology of SARS-CoV-2 in Seychelles, 2020–2021

**DOI:** 10.3390/v14061318

**Published:** 2022-06-16

**Authors:** John Mwita Morobe, Brigitte Pool, Lina Marie, Dwayne Didon, Arnold W. Lambisia, Timothy Makori, Khadija Said Mohammed, Zaydah R. de Laurent, Leonard Ndwiga, Maureen W. Mburu, Edidah Moraa, Nickson Murunga, Jennifer Musyoki, Jedida Mwacharo, Lydia Nyamako, Debra Riako, Pariken Ephnatus, Faith Gambo, Josephine Naimani, Joyce Namulondo, Susan Zimba Tembo, Edwin Ogendi, Thierno Balde, Fred Athanasius Dratibi, Ali Ahmed Yahaya, Nicksy Gumede, Rachel A. Achilla, Peter K. Borus, Dorcas W. Wanjohi, Sofonias K. Tessema, Joseph Mwangangi, Philip Bejon, David J. Nokes, Lynette Isabella Ochola-Oyier, George Githinji, Leon Biscornet, Charles N. Agoti

**Affiliations:** 1Kenya Medical Research Institute-Wellcome Trust Research Programme (KWTRP), Kilifi P.O. Box 230-80108, Kenya; alambisia@kemri-wellcome.org (A.W.L.); tmakori@kemri-wellcome.org (T.M.); ksaid@kemri-wellcome.org (K.S.M.); zdelaurent@gmail.com (Z.R.d.L.); lndwiga@kemri-wellcome.org (L.N.); wmburu@kemri-wellcome.org (M.W.M.); emoraa@kemri-wellcome.org (E.M.); nmurunga@kemri-wellcome.org (N.M.); jmusyoki@kemri-wellcome.org (J.M.); jmwacharo@kemri-wellcome.org (J.M.); lnyamako@kemri-wellcome.org (L.N.); driako@kemri-wellcome.org (D.R.); pephantus@kemri-wellcome.org (P.E.); fgambo@kemri-wellcome.org (F.G.); jnaimani@kemri-wellcome.org (J.N.); jmwangangi@kemri-wellcome.org (J.M.); pbejon@kemri-wellcome.org (P.B.); jnokes@kemri-wellcome.org (D.J.N.); liochola@kemri-wellcome.org (L.I.O.-O.); ggithinji@kemri-wellcome.org (G.G.); cnyaigoti@kemri-wellcome.org (C.N.A.); 2Seychelles Public Health Laboratory, Public Health Authority, Ministry of Health, Victoria P.O. Box 52, Seychelles; brigitte.pool@health.gov.sc (B.P.); l.marie@health.gov.sc (L.M.); dwayne.didon@health.gov.sc (D.D.); leon.biscornet@health.gov.sc (L.B.); 3World Health Organization-Seychelles Country Office, Victoria P.O. Box 1217, Seychelles; namulondoj@who.int (J.N.); tembosu@who.int (S.Z.T.); ogendie@who.int (E.O.); 4Department of Arbovirology, Emerging and Re-emerging Infectious Diseases—Uganda Virus Research Institution (UVRI), Entebbe P.O. Box 49, Uganda; 5World Health Organization Regional Center for Africa, Brazzaville P.O. Box 06, Congo; baldet@who.int (T.B.); dratibif@who.int (F.A.D.); aliahmedy@who.int (A.A.Y.); gumedemoeletsih@who.int (N.G.); achillar@who.int (R.A.A.); 6World Health Organization-Kenya Country Office, Gigiri, Nairobi P.O. Box 45335, Kenya; borusp@who.int; 7Africa Centres for Disease Control and Prevention (Africa CDC), Addis Ababa P.O. Box 3243, Ethiopia; dorcasw@africa-union.org (D.W.W.); sofoniast@africa-union.org (S.K.T.); 8School of Life Sciences and Zeeman Institute for Systems Biology and Infectious Disease Epidemiology Research (SBIDER), University of Warwick, Coventry CV4 7AL, UK; 9Department of Biochemistry and Biotechnology, Pwani University, Kilifi P.O. Box 195-80108, Kenya; 10Department of Public Health, Pwani University, Kilifi P.O. Box 195-80108, Kenya

**Keywords:** Seychelles, SARS-CoV-2, variants of concern

## Abstract

Seychelles, an archipelago of 155 islands in the Indian Ocean, had confirmed 24,788 cases of severe acute respiratory syndrome coronavirus 2 (SARS-CoV-2) by the 31st of December 2021. The first SARS-CoV-2 cases in Seychelles were reported on the 14th of March 2020, but cases remained low until January 2021, when a surge was observed. Here, we investigated the potential drivers of the surge by genomic analysis of 1056 SARS-CoV-2 positive samples collected in Seychelles between 14 March 2020 and 31 December 2021. The Seychelles genomes were classified into 32 Pango lineages, 1042 of which fell within four variants of concern, i.e., Alpha, Beta, Delta and Omicron. Sporadic cases of SARS-CoV-2 detected in Seychelles in 2020 were mainly of lineage B.1 (lineage predominantly observed in Europe) but this lineage was rapidly replaced by Beta variant starting January 2021, and which was also subsequently replaced by the Delta variant in May 2021 that dominated till November 2021 when Omicron cases were identified. Using the ancestral state reconstruction approach, we estimated that at least 78 independent SARS-CoV-2 introduction events occurred in Seychelles during the study period. The majority of viral introductions into Seychelles occurred in 2021, despite substantial COVID-19 restrictions in place during this period. We conclude that the surge of SARS-CoV-2 cases in Seychelles in January 2021 was primarily due to the introduction of more transmissible SARS-CoV-2 variants into the islands.

## 1. Introduction

Seychelles, an archipelago of 155 islands in the Indian Ocean with a population size of approximately 99,202 [1], confirmed its first cases of severe acute respiratory syndrome coronavirus 2 (SARS-CoV-2), the etiological agent of coronavirus disease 2019 (COVID-19), on the 14th of March 2020. This was shortly after the World Health Organisation (WHO) declared COVID-19 a global pandemic on the 11th of March 2020. However, the number of COVID-19 cases in Seychelles remained low (average of 1 case/day) until January 2021 when a surge was observed in the country (average of 72 cases/day). As of 31 December 2021, Seychelles had reported 24,788 laboratory-confirmed COVID-19 cases, >98% of which were recorded in 2021 [2]. The surge of the number of COVID-19 cases in Seychelles in 2021 could be due to two major factors: (a) the relaxation of government COVID-19 stringency measures and (b) the arrival of more transmissible SARS-CoV-2 variants on the islands. Our analysis looked at these factors in an attempt to improve understanding of the COVID-19 transmission dynamics in Seychelles.

Various COVID-19 countermeasures were announced periodically by the Seychelles government to curb further introduction and spread of the virus following first detection on 14 March 2020. These measures included: a 14-day quarantine for people returning from countries with significant COVID-19 community transmission on the 16 March 2020, closure of day care centres and learning institutions, and ban on international arrivals and foreign travel by Seychellois citizens except for medical emergencies beginning 23 March 2020, a 21-day nationwide lockdown, tracing, isolation and monitoring of all persons who had close contact with COVID-19 patients for 14 days beginning 6 April 2020, closure of all shops except those that sell food items, groceries or pharmaceutical products beginning 6 April 2020, workplace closures and restriction of outdoor movement except for essential services on 9 April.

With the countermeasures seeming to work in minimizing COVID-19 cases on the islands, on the 4 of May 2020, the Seychelles government eased some of the COVID-19 restrictions, including opening of all day care and learning institutions, opening of all shops, and lifting of the ban on restrictions of movement of people. In June 2020, the Seychelles government lifted the ban on international travel and allowed visitors (international tourists) from countries categorised as low-risk, but with a requirement to show a COVID-19 negative certificate (RT-PCR test). Despite this removal of many of the Government restrictions, the number of SARS-CoV-2 cases in Seychelles throughout 2020 remained low.

Towards the end of 2020 and in 2021, in widely different geographical locations, five SARS-CoV-2 variants of concern (VOC)—Alpha, Beta, Gamma, Delta and Omicron; five variants of variants of interest (VOI)—Eta, Kappa, Iota, Epsilon and Theta; and over 10 variants under monitoring (VUM) emerged that appeared to be considerably more transmissible and with the potential to escape pre-existing immunity [3,4]. Notably, soon after their emergence, a surge of COVID-19 cases was observed in Seychelles in early 2021. Further, in the last quarter of 2021, Omicron SARS-CoV-2 variants of concern (VOCs) were detected globally [4,5]. The objective of this study was to describe the genomic epidemiology of SARS-CoV-2 in Seychelles and in particular lineages coinciding with the surge of cases that began in January 2021, with the aim of improving understanding of the introduction and transmission of SARS-CoV-2 in Seychelles.

## 2. Materials and Methods

### 2.1. Ethical Statement

The SARS-CoV-2 positive samples were sequenced at the Kenya Medical Research Institute (KEMRI) Wellcome Trust Research Programme (KWTRP) as part of a regional collaborative COVID-19 public health rapid response initiative overseen by WHO-AFRO and Africa-CDC. KWTRP Kilifi is one of the 12 designated WHO-AFRO/Africa-CDC regional reference laboratories for SARS-CoV-2 genomic surveillance in Africa. The whole genome sequencing study protocol was reviewed and approved by the Scientific and Ethics Review Committee (SERU) at KEMRI, (SERU #4035). Individual patient consent requirement was waivered by the committee as the sequenced samples were part of the public health emergency response.

### 2.2. Study Site and Samples

A total of 1298 SARS-CoV-2 real-time polymerase chain reaction (qRT-PCR)-confirmed nasopharyngeal and oropharyngeal (NP/OP) positive swab samples collected between 14 March 2020 and 31 December 2021 were targeted for whole genome sequencing. The samples received for sequencing at KWTRP for whole genome sequencing were selected considering cycle threshold (Ct) value cut off <30. The monthly temporal distribution of samples selected for whole genome sequencing is shown in Appendix A.

### 2.3. Laboratory Procedures

#### 2.3.1. RNA Extraction, cDNA Synthesis and Amplification

The NP/OP swab samples on arrival at KWTRP laboratories were re-extracted using the QIAamp Viral RNA Mini Kit (Qiagen, Manchester, UK) following the manufacturer’s instructions, starting at volume 140 µL, and elution volume of 60 μL. The RNA was then re-assayed to confirm SARS-CoV-2 genetic material using one of three commercial kits, namely Da An Gene Co. Ltd.’s Detection Kit (Guangzhou, China) (targeting N gene or ORF1ab), SD Biosensor’s Standard M Real Time Detection Kit (South Korea) (targeting E gene and ORF1ab) and KH Medical’s RADI COVID-19 Detection Kit (South Korea) (targeting RdRp and S genes), while following manufacturer’s instructions. Samples with Ct values <33 were selected for cDNA synthesis.

Extracted RNA was reverse transcribed using the LunaScript^®^ RT SuperMix Kit (New England Biolabs, Ipswich, MA, USA). For each of the selected samples, 2 μL of LunaScript^®^ RT SuperMix was added to 8 μL of RNA template, incubated at 25 °C for 2 min, 55 °C for 10 min, held at 95 °C for 1 min and placed on ice for 1 min. The generated viral cDNA was amplified using the Q5^®^ Hot Start High-Fidelity 2× Master Mix (NEB, Ipswich, MA, USA) along with ARTIC nCoV-2019 version 3 primers (primer pools A and B), as documented previously [6]. The thermocycling conditions involved a touchdown PCR with the following conditions: heat activation at 98 °C for 30 s, followed by 40 amplification cycles (i.e., 25 cycles of 98 °C for 15 s and 65 °C for 5 min, and 15 cycles of 62.5 °C for 5 min and 98 °C for 15 s and one cycle at 62.5 °C for 5 min), final extension at 62.5 °C for 5 min, followed by a final hold at 4 °C. To overcome amplicon dropouts in regions 3, 9, 17, 26, 64, 66, 67, 68, 74, 88, 91 and 92 of the genomes [6], primer pairs for the aforementioned regions were constituted in an additional pool, named herein pool C. After the multiplex PCR, an agarose gel electrophoresis step was included to exclude samples with no visible bands from further processing.

#### 2.3.2. Oxford Nanopore Library Preparation and Sequencing

For each sample, the PCR products of primer pools A, B and C were combined to make a total of 23 μL (all of pool A (10 μL), pool B (10 μL) and pool C (3 µL)) and cleaned using 1× AMPure XP beads (Beckman Coulter, Indianapolis, IND, USA), followed by two ethanol (80%) washes. The pellet was resuspended in 20 μL nuclease-free water and 1 μL of the eluted sample was quantified using the Qubit dsDNA HS Assay Kit (ThermoFisher Scientific, San Francisco, CA, USA). End-prep reaction was performed according to the ARTIC nCoV-2019 sequencing protocol v3 (LoCost) with 200 fmol (50 ng) of amplicons and the NEBNext Ultra II End repair/dA-tailing Kit (NEB, Massachusetts, USA) and incubated at 20 °C for 5 min and 65 °C for 5 min. From this, 1 μL of DNA was used for barcode ligation using Native Barcoding Expansion 96 (Oxford Nanopore Technology, Oxford, UK) and NEBNext Ultra II Ligation Module reagents (NEB, Massachusetts, USA). Incubation was performed at 20 °C for 20 min and at 65 °C for 10 min. This step was eventually modified to employ NEBNext Blunt/TA Ligase Master Mix (NEB, Massachusetts, USA) using the same barcodes and incubation conditions.

The barcoded samples were pooled together. The pooled and barcoded DNA samples were cleaned using 0.4× AMPure XP beads followed by two ethanol (80%) washes and eluted in 1/14 of the original volume of nuclease-free water. Adapter ligation was performed using 50–100 ng of the barcoded amplicon pool, NEBNext Quick Ligation Module reagents (NEB, Massachusetts, USA) and Adaptor Mix II (ONT, Oxford, UK), and incubated at room temperature for 20 min. Final clean-up was performed using 1× AMPure XP beads and 125 μL of Short Fragment Buffer (ONT, Oxford, UK). The library was eluted in 15 μL Elution Buffer (ONT, Oxford, UK). The final library was normalised to 15–70 ng, loaded on a SpotON R9 flow cell and sequenced on a MinION Mk1B or GridION device [6].

#### 2.3.3. SARS-CoV-2 Genome Consensus Assembly

The data generated via the MinION and GridION devices were processed using the ARTIC bioinformatic protocol (https://artic.network/ncov-2019/ncov2019-bioinformatics-sop.html) (accessed on 2 October 2020). In brief, raw FAST5 files were base called and demultiplexed using ONT’s Guppy v4.0.5in high accuracy mode using a minimum Q score of 7. FASTQ reads between 300 bps and 750 bps were filtered using the upcycle module. The consensus sequences were generated by aligning base called reads against the SARS-CoV-2 reference genome (GenBank accession MN908947.3) using MiniMap2 [7]. All positions with a genome coverage of less than 20 reads were masked with Ns. The consensus sequences were then polished using Nanopolish toolkit (version 0.13.3) using the raw signals.

#### 2.3.4. Lineage and VOC Assignment

Additional quality control, clade assignment and mutation profiles were obtained using the NextClade tool v1.13.2 [8] using a SARS-CoV-2 reference genome (accession NC_045512). All consensus sequences with a genome coverage >70% were classified using the PANGO lineage assignment tool (Pangolin v3.1.20 and PangoLearn v02.02.2022) [9].

#### 2.3.5. Global Comparison Sequences

Seychelles sequences were analysed against a backdrop of globally representative SARS-CoV-2 lineages. To ensure global representation of sequences, we downloaded all the sequences (*n* = 8,916,634) from the Global Initiative on samples. Sharing All Influenza Data (GISAID) database collected before 31 December and used an in-house R script to randomly select a sub-sample of 5179 genomes while considering Pango lineage (lineages detected in Seychelles only), continent and date of collection. These global random genomes were collected from 150 countries and territories between 2 May 2020 and 31 December 2021.

#### 2.3.6. Phylogenetic Reconstruction

The retrieved global sequence dataset, the sequences from Seychelles, were aligned using Nextalign version 1.4.1 (https://github.com/neherlab/nextalign) (accessed on 29 March 2022). against the reference SARS-CoV-2 genome (accession NC_045512). A maximum likelihood (ML) phylogeny was inferred using IQTREE version 2.1.3 (http://www.iqtree.org/) (accessed on 29 March 2022). The software initiates tree reconstruction after assessment and selection of the best model of nucleotide substitution for the alignment. TreeTime v0.8.1 [10] was used to transform the ML tree topology into a time calibrated phylogenetic tree. The resulting trees were visualized using the Bioconductor ggTree v2.2.4 package [11] in R [12].

#### 2.3.7. Estimation of Virus Importation and Exportation into Seychelles

The global ML tree topology was used to estimate the number of viral transmission events between Seychelles and the rest of the world as described previously [13,14]. Briefly, TreeTime was used to transform the ML tree topology into a dated phylogenetic tree, mapping the location of sampled sequences to the external tips of the trees. Outlier sequences (*n* = 208) were identified by TreeTime and excluded during this process. The migration model of TreeTime also infers the most likely location for internal nodes in the trees. We then count3d the number of state changes from the root to the external tips. The state changes are counted when an internal node transitions from one country to a different country in the resulting child-node or tip(s). The timing of transition events is then recorded which serves as the estimated import or export event. To validate our estimates, we conducted the analysis with two different sets of data randomly sampled from GISAID.

#### 2.3.8. Statistical Analysis

All statistical analyses were performed using R v4.1.0 [12].

## 3. Results

### 3.1. Sequenced COVID-19 Cases in Seychelles

The rise of COVID-19 cases in 2021 was preceded by a period of relaxed countermeasures to curb the spread of the virus (Figure 1A,B). The roll-out of vaccine in the country appeared to have no effect on the number of COVID-19 cases reported in the country i.e., we saw surge of cases due to Delta VOC in May–June 2021, when at least 60% of the population had received their first dose of the vaccine (Figure 1B–D). Of 1298 SARS-CoV-2 positive samples received at KWTRP for genome sequencing, near complete genomes (>70% genome coverage) were recovered from 1056 samples, and these were used in the subsequent lineage and phylogenetic analysis (Appendix A). A summary of the demographic details for the samples successfully sequenced and those that failed are provided in Table 1.

### 3.2. SARS-CoV-2 Lineages Circulating in Seychelles

The recovered 1056 genomes were classified into 32 distinct Pango lineages, 28 of which occurred within VOC, VOI or VUM: Alpha VOC (*n* = 1), Beta VOC (*n* = 1), Delta VOC (*n* = 21) and Omicron VOC (*n* = 3) and Kappa VOI (*n* = 1) and B.1.640.2 VUM (*n* = 1) (Table 2). A total of four non-VOC/non-VOI/non-VUM lineages were detected among the sequenced infections in Seychelles: B.1 (*n* = 9), B.1.1 (*n* = 1), B.1.1.50 (*n* = 1) and lineage B.1.1.277 (*n* = 1). Lineage B.1 (predominantly detected in Europe) was the first lineage to be detected in Seychelles, in a sequenced sample from June 2020, followed by lineage B.1.1.277 (predominantly detected in Europe) in October 2020, B.1.1.50 (predominantly detected in Israel and Palestine) in January 2021 followed by B.1.1 in May 2021. The non-VOC/VOI lineages were replaced by the Beta VOC (B.1.351) in February 2021, which was later subsequently replaced by the Delta VOC in May 2021 that dominated until November 2021 when Omicron cases were first identified (Figure 1D and Figure 2).

Detection of Beta VOC in February 2021 coincided with the start of a surge of COVID-19 cases, with a further sharp surge observed in May 2021 coinciding with detection of Delta VOC in May 2021 (Figure 1B,D). Since the emergence of Delta VOC in Seychelles in May 2021, a total of 21 Delta VOC lineages co-circulated with varying frequency. Lineage AY.122 (*n* = 742) was the most prevalent, with detections until the end of the surveillance period covered in this analysis (Figure 1D and Figure 2). Other common Delta lineages were AY.43 (*n* = 33) and B.1.617.2 (*n* = 13). We observed the start of another surge in November 2021 due to Omicron (lineages BA.1 (*n* = 126), BA.1.1 (*n* = 18) and BA.2 (*n* = 1) (Figure 1B,F and Appendix A) which peaked in mid-January 2022 (not shown).

### 3.3. Phylogenetic Clustering of Seychelles Sequences

Genetic distance-resolved phylogeny of the Seychelles genomes, including global reference sequences (*n* = 5179), revealed that most of the Seychelles sequences were interspersed as clusters (>2 sequences) or singletons across the phylogenetic trees, suggesting multiple viral introductions into Seychelles (Figure 3). In the VOC/non-VOC-specific phylogenies, Delta VOC (*n* = 863) grouped into 14 clusters (>2 sequences) and 12 singletons on the global phylogenetic tree pointing to separate introductions of the Delta VOC into the country, whereas Omicron VOC (*n* = 145) clustered into 10 clusters and 11 singletons (Figure 3D,E). Seychelles’ Beta (*n* = 29) and Alpha (*n* = 5) viruses clustered closely amongst themselves suggesting few introductions that led to onward transmission in Seychelles (Figure 3B,C). Seychelles B.1 viruses sampled in 2020 were dispersed on the global phylogeny as singletons, most likely pointing to multiple introductions into Seychelles during the initial phase of the pandemic (Figure 3A). Sequences from different locations (i.e., districts) in Seychelles clustered closely or together on the phylogenetic tree, a feature suggesting rapid spread of the virus within the country over a short period of time (not shown).

### 3.4. SARS-CoV-2 Diversity and Mutational Analysis in Seychelles Genomes

SARS-CoV-2 variant analysis comparing sequences from Seychelles to the Wuhan reference sequence (NC_045512.2) detected a total of 703 amino acid mutations across different gene regions. We identified a total of 27 amino acid mutations and two deletions that had a prevalence of >50% in all the sequenced cases (Appendix A). The most prevalent amino acid mutation was D614G (A23403G) (98.9%) occurring in the spike glycoprotein, followed by P314L (C14408T) (93.3%) in the open reading frame 1b (ORF1b) (Appendix A). Of the 32 lineages detected in Seychelles, Omicron, and Delta VOC were the most evolved, with the highest genetic diversity (Appendix A). The number of mutations in these variants varied from sample to sample, with Omicron having a mean of 58 (range of 38–69) mutations in the majority of the genomes, while Delta presented a mean of 34 (range of 17–57) mutations in the majority of the genomes (Appendix A).

### 3.5. Export and Import of SARS-CoV-2 Lineages in Seychelles

Ancestral location state reconstruction of the dated global phylogeny (Figure 4A) was used to infer the number of viral importations and exportations. In total, between the 25th of June 2020 and the 31 December 2021, we inferred at least 78 importations into Seychelles with 28 (35%) of the introductions coming from Europe, 21 separate introductions from Africa, 15 separate introductions from Asia, six introductions from North America, five separate introductions from Oceania and three introductions from South America. (Figure 4B). Of the 78 detected viral imports into Seychelles, 66 occurred between January and December 2021 after the rise in COVID-19 cases was experienced in the Seychelles (Figure 4C). From the analysis, we also inferred 32 export events from Seychelles to the rest of the world, mainly Asia (*n* = 10) Europe (*n* = 8) and Africa (*n* = 6). The re-analysis using different set of sub-samples found results that were closely aligned with those revealed by sub-sample one, and thus similar conclusions (Appendix A).

## 4. Discussion

We estimated at least 78 independent introductions of SARS-CoV-2 into Seychelles between 25 June 2020 to 31 December 2021, with the importations likely originating from all the continents in the world. Notably, the surge of COVID-19 cases from January to December 2021 was characterised by detections of VOCs in the country. The Beta variant of concern was the dominant strain in circulation from February to April 2021 and probably responsible for the surge in cases in January 2021 (highest daily number of infections at 231, on the 4 of March 2021), but the introduction and rapid spread of the Delta variant seen from May to December 2021 caused a further surge (highest daily number of infections at 484, on the 6 of May 2021).

The low number of cases detected for non-VOCs (B.1, B.1.1, B.1.1.50 and B.1.1.277) lineages in 2020, could be explained by the fact that this insular population experienced very low importations due to the COVID-19 restriction countermeasures that were in place in 2020 but also that the non-VOCs were not highly transmissible. We ascertain that the introduction of the VOCs with high transmissibility was probably the cause of the surge of cases in Seychelles in 2021. However, these assertions should be taken with caution since extensive genomic surveillance in the country began in early 2021. To note, the detection of VOCs in the country preceded the period of relaxed COVID-19 restriction countermeasures, including allowing tourists into the country and the resumption of in person school classes. This may have led to increased viral importations, as detected in our dataset, and to spread in the country.

The detection of Delta VOCs in May 2021 coincided with a period of relaxed COVID-19 restriction countermeasures (Oxford stringency index <50) and the surge of cases during this time may have been due to multiple introductions and rapid spread of the virus in the country. By December, a total of 21 different Delta lineages were detected in Seychelles. The first Omicron detection in Seychelles was on the 29th of November 2021. By 31 December 2021, three Omicron lineages had been detected in Seychelles including BA.2 which has been noted to be rapidly growing across the world [16,17,18].

Unsurprisingly, the majority of the Seychelles sequences harbored important spike mutations e.g., D614G mutation occurring in 98.2% of all the sequenced cases. The D614G amino acid change has been associated with stronger interaction between the virus and the angiotensin-converting enzyme 2 (ACE2) affinity, leading to higher infectivity and transmissibility [19,20], Other important mutations included: (i) Beta VOC; K417N, E484K and N501Y occurring in the spike receptor binding domain (RBD), which have been associated with reduced sensitivity to convalescent and post-vaccination sera [21]; (ii) Delta VOC, L452R, and P681R mutations, which have been linked to reduced sensitivity to neutralizing antibodies and higher transmissibility [22,23]; (iii) Alpha VOC; N501Y and P681H mutations and (iv) Omicron VOC: Q498R and N501Y occurring in the RBD have been linked to ACE2 binding affinity [24]. These RBD mutations coupled with four amino acid substitutions (i.e., A67V, T95I, and L212) and three deletions (67–70, 142–144 and 211) and an insertion (EPE at position 214) in the N-terminal domain (NTD) are linked to reduced sensitivity to convalescent and post-vaccination sera [18,25]. A cluster of three mutations occurring near the S1–S2 furin cleavage site (H655Y, N679K and P681H) have been associated with increased transmissibility [26].

Our phylogenetic analysis showed that Seychelles sequences virus diversity was nested within the global virus diversity (i.e., Seychelles sequences clustered with sequences sampled from different countries, suggestive of global spread of SARS-CoV-2 lineages). The close association between the viruses and those from other countries reflects global transmission of the virus as a result of global migration, increased connectivity, and social mixing. Further, focusing on the VOCs, we observed patterns of SARS-CoV-2 viral diversity inside Seychelles; phylogenetic clusters consisted of viruses which were derived from different geographic locations and formed a deep hierarchical structure, indicating an extensive and persistent nation-wide transmission of the virus.

Our findings are consistent with findings from island countries such as Comoros, Reunion and New Zealand; these countries were able to contain the first pandemic wave starting in March 2020, due to COVID-19 strict countermeasures such as the ban on international arrivals, which may have led to limited viral introduction into the islands, or perhaps viral introductions during the early phase of the pandemic did not result in community transmission due to government countermeasures against COVID-19 such as countrywide lockdown or self-isolation of the entire population [27,28]. Surges of COVID-19 cases in island populations appear to be majorly driven by introductions of VOCs. For example, Comoros seems to have experienced its first surge of COVID-19 cases after introduction of Beta VOCs into the population in January 2021 [29], similar to the period when Seychelles saw its first surge, also due to Beta VOC.

This study had some limitations. First, our import/export inferences can be influenced by sampling biases of the global dataset and the low rate of sequencing in Seychelles. Therefore, the true number of international introductions is likely significantly higher than that reported here. Second, incomplete metadata for some samples limited the scope of our analysis, for example, lack of location of samples collected disallowed investigation of the transmission pathway of viruses within the country. Third, SARS-CoV-2 sequences from Seychelles are only available from a very small fraction of the number of confirmed cases into the country.

These data reinforce the importance of genomic surveillance in Seychelles as a tool for monitoring and providing real-time information on the spread of emerging SARS-CoV-2 variants in the population with important implications for public health and immunization strategies. The surge of COVID-19 cases due to VOCs during a period of heightened COVID-19 countermeasures raises questions on the optimal timing of the introduction of public health interventions. When the interventions are introduced after a surge has started, it is often too late, and the control strategies should focus on local transmission to understand characteristics and origins of locally circulating SARS-CoV-2 diversity in order to prevent further spread [14]. Moreover, studies on genomic surveillance would also be useful in investigating vaccine effectiveness against circulating variants which appear to have a high turnover.

## Figures and Tables

**Figure 1 viruses-14-01318-f001:**
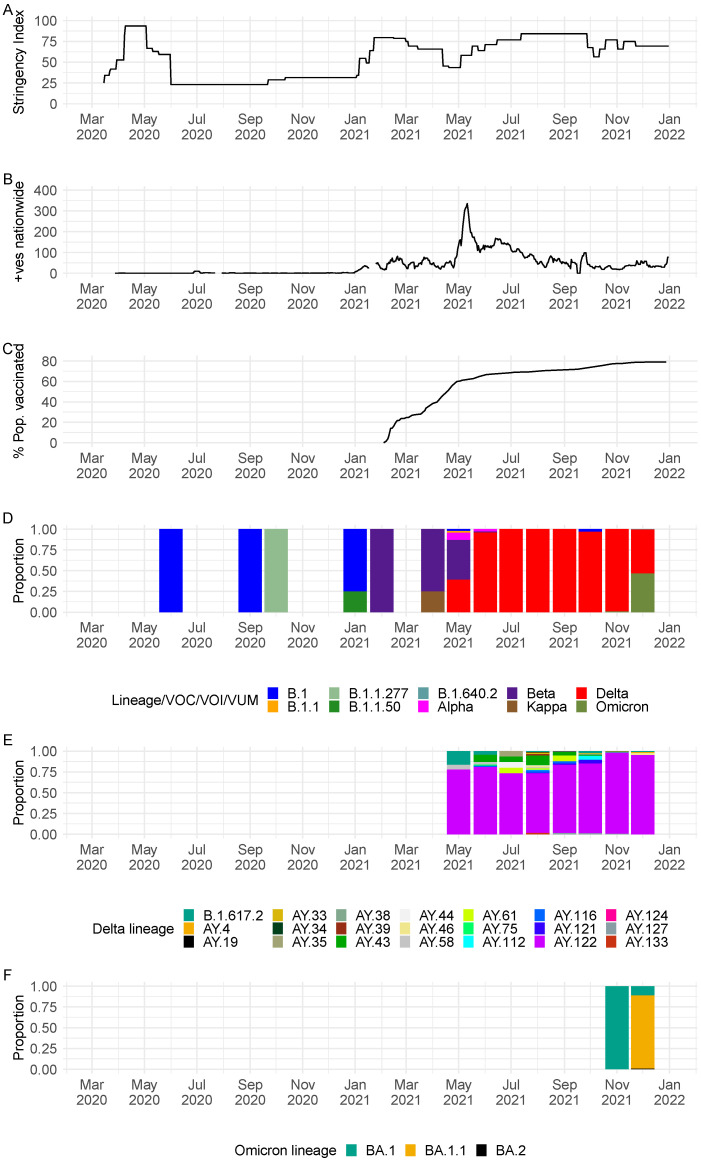
(**A**) Seychelles government intervention levels as measured by the Oxford stringency index [15]. (**B**) An epidemic curve for Seychelles derived from the daily positive case numbers obtained from https://ourworldindata.org/coronavirus/country/seychelles (accessed on 5 May 2022). (**C**) Percentage of the population administered with vaccine; data obtained from https://ourworldindata.org/coronavirus/country/seychelles (accessed on 5 May 2022). (**D**) Monthly temporal pattern of SARS-CoV-2 lineages and variants in Seychelles among the 1056 samples sequenced from COVID-19 positive cases from the Seychelles (25 June 2020, to 31 December 2021). (**E**) Monthly temporal distribution of Delta VOC lineages among samples sequenced from COVID-19 positive cases from the Seychelles (25 June 2020, to 31 December 2021). (**F**) Monthly temporal distribution of Omicron VOC lineages among samples sequenced from COVID-19 positive cases from the Seychelles (25 June 2020, to 31 December 2021).

**Figure 2 viruses-14-01318-f002:**
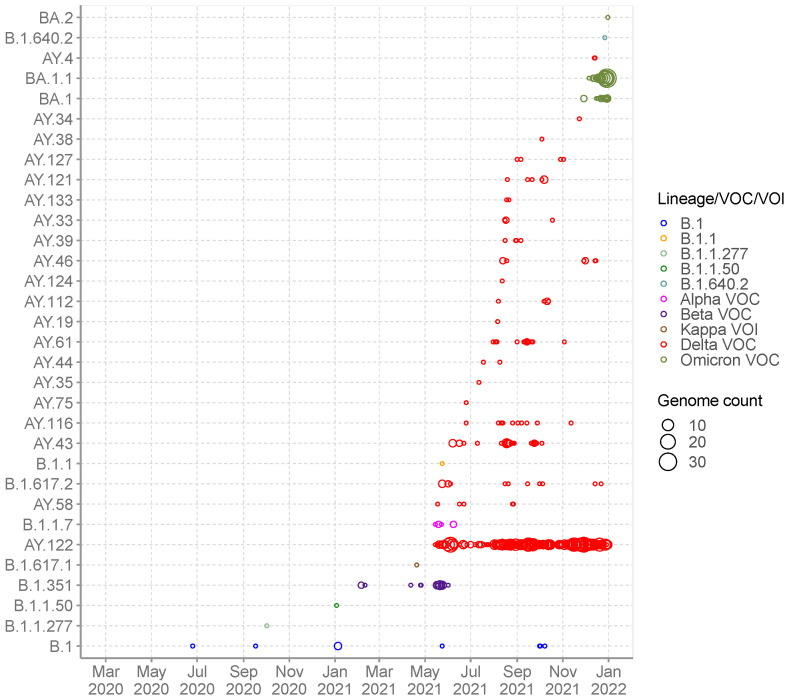
SARS-CoV-2 Pango lineages in the sequenced 1056 Seychelles samples and timing of detections (circle size scaled by number of daily detections).

**Figure 3 viruses-14-01318-f003:**
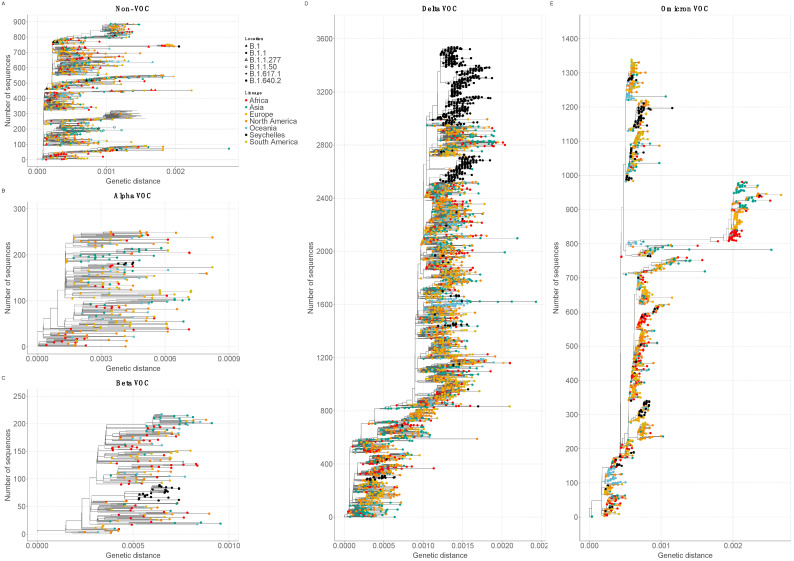
Genetic distance-resolved lineage-specific phylogenetic trees for Omicron, Alpha, Beta, Delta VOC, and Non-VOC. Seychelles genomes are indicated with colored tip labels. (**A**) Phylogeny of Non-VOC that combined 14 Seychelles sequences and 875 global sequences. (**B**) Phylogeny of Alpha VOC that combined 5 Seychelles sequences and 246 global sequences. (**C**) Phylogeny of Beta VOC that combined 29 Seychelles sequences and 187 global sequences. (**D**) Phylogeny of Delta VOC that combined 863 Seychelles sequences and 2676 global sequences. (**E**) Phylogeny of Omicron VOC that combined 145 Seychelles sequences and 1195 global sequences.

**Figure 4 viruses-14-01318-f004:**
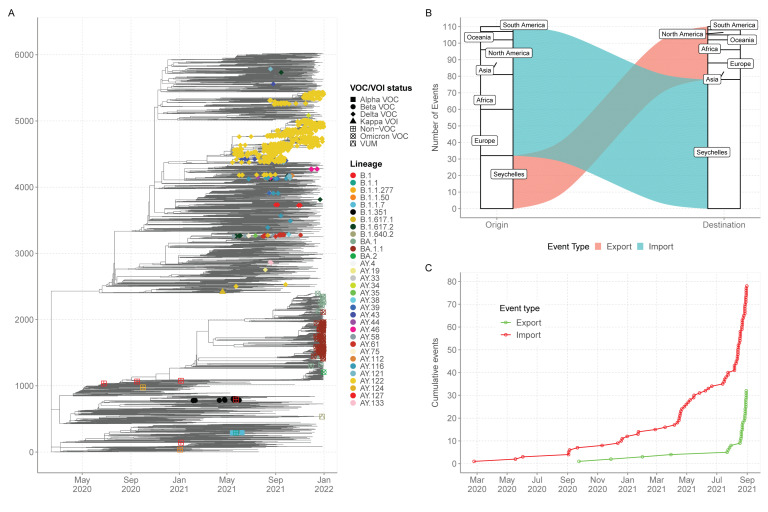
(**A**) Time-resolved global phylogeny that combined 1056 Seychelles sequences (coloured tip labels) and 5179 global reference sequences. (**B**) The number of viral imports and exports into and out of Seychelles. (**C**) Cumulative number of viral imports and export over time into Seychelles.

**Table 1 viruses-14-01318-t001:** Demographic characteristics of SARS-CoV-2 positive samples received for sequencing at KWTRP. Sample were collected between 14 March 2020 and 31 December 2021 (*n* = 1298).

	Sample Sequenced		Samples Not Sequenced		Total
	Number (*n* = 1056)	Population Proportion (%)	Number (*n* = 242)	Population Proportion (%)	
**Sex**					
Female	560	53.0	120	49.6	680
Male	473	44.8	102	42.1	575
Unknown	23	2.2	20	8.3	43
**Age**					
Mean	33.4 (18.3)	-	34.3 (20.4)	-	
Median	32	-	34	-	
Min, Max	0, 98	-	0, 89	-	
Missing	18	1.7	9	3.7	27
**Age distribution**					
0–9	106	10.0	37	15.3	143
10–19	132	12.5	28	11.6	160
20–29	215	20.4	30	12.4	245
30–39	225	21.3	42	17.4	267
40–49	146	13.8	45	18.6	191
50–59	127	12.0	21	8.7	148
60–69	58	5.5	19	7.9	77
70–79	14	1.3	6	2.5	20
>80	15	1.4	5	2.1	20
**Travel information**					
Yes	3	0.3	8	3.3	11
No	1053	99.7	234	96.7	1287
**Symptoms**					
Asymptomatic	37	3.5	31	12.8	68
Symptomatic	273	25.9	52	21.5	325
Deceased	3	0.3	4	1.7	7
Missing	738	69.9	155	64.0	893

**Table 2 viruses-14-01318-t002:** Description of SARS-CoV-2 lineages observed in Seychelles, their global history and VOC/VOI status.

Non-VOC/VOI/VOC/VUM	Lineage	Freq	Proportion (%)	Earliest Date	Description
**Non-VOC/VOI**	B.1	9	0.9	1 January 2020	Predominantly found in Europe
	B.1.1	1	0.1	1 January 2020	Predominantly found in Europe
	B.1.1.277	1	0.1	7 March 2020	Predominantly found in Europe
	B.1.1.50	1	0.1	29 March 2020	Predominantly found in Israel and Palestine
**VUM**	B.1.640.2	1	0.1	15 October 2021	Predominantly found in France
**Kappa VOI**	B.1.617.1	1	0.1	3 March 2020	Kappa variant of interest, predominantly found in India lineage with 484Q
**Alpha VOC**	B.1.1.7	5	0.5	7 February 2020	Alpha variant of concern, first identified in UK
**Beta VOC**	B.1.351	29	2.7	27 March 2020	Beta variant of concern, first identified in South Africa
**Delta VOC**	B.1.617.2	13	1.2	15 April 2020	Predominantly found in India
	AY.4	2	0.2	3 August 2020	Predominantly found in UK
	AY.19	1	0.1	7 April 2021	Predominantly found in South Africa
	AY.33	4	0.4	13 June 2020	Lineage circulating mostly in Belgium, Denmark, France, Netherlands, Germany
	AY.34	1	0.1	18 November 2020	Predominantly found in UK
	AY.35	1	0.1	21 August 2020	Predominantly found in lineage with spike E484Q circulating in USA
	AY.38	1	0.1	27 March 2021	Predominantly found in in South Africa
	AY.39	4	0.4	14 January 2021	Predominantly found in USA
	AY.43	33	3.1	21 August 2021	Predominantly found in European
	AY.44	2	0.2	11 May 2020	Predominantly found in USA
	AY.46	8	0.8	15 October 2021	Predominantly found in Africa
	AY.58	5	0.5	16 March 2021	Predominantly found in Italy
	AY.61	15	1.4	7 January 2021	Predominantly found in Italy
	AY.75	1	0.1	6 January 2021	Predominantly found in USA
	AY.112	5	0.5	5 December 2020	Predominantly found in India
	AY.116	11	1.0	21 January 2021	Africa lineage
	AY.121	7	0.7	24 January 2021	Predominantly found in Turkey
	AY.122	742	70.3	7 September 2020	European lineage
	AY.124	1	0.1	9 January 2021	Predominantly found in Portugal and other European countries
	AY.127	4	0.4	10 December 2020	Predominantly found in India
	AY.133	2	0.2	10 February 2021	Predominantly found in France
**Omicron VOC**	BA.1	18	1.7	10 September 2021	Predominantly found in UK
	BA.1.1	126	11.9	13 September 2021	Predominantly found in USA
	BA.2	1	0.1	17 November 2021	Predominantly found in UK

## Data Availability

All data generated and analysis script for this manuscript are available from the Virus Epidemiology and Control, Kenya Medical Research Institute (KEMRI)–Wellcome Trust Research Programme data server, https://doi.org/10.7910/DVN/AYT2UA (accessed on 29 March 2022). Whole genome sequences are available from GISAID database accession number EPI_ISL_4880527-EPI_ISL_4880796, EPI_ISL_5942854-EPI_ISL_6705093, EPI_ISL_8424404-EPI_ISL_8424612 and EPI_ISL_11060102-EPI_ISL_11060407.

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
