# Peer review of "Genomic Epidemiology of SARS-CoV-2 in Seychelles, 2020–2021"

_viruses, 2022, doi:10.3390/v14061318_

Round 1

Reviewer 1 Report

The manuscript presents an analysis of the genomic epidemiology of SARS-CoV-2 in the Seychelles archipelago based on 1056 complete genome sequences sampled between March 2020 and December 2021. This is very similar work to many other publications on the same topic for different countries, regions, or even cities and as such is an interesting contribution to the global understanding of the introduction and spread of an important emergent virus. However, I think that the work can be improved before it is accepted for publication. My main points for improvement are detailed next.

The procedure for estimating the number of virus importations into Seychelles needs to be explained in more detail. According to the current description, a sequence of transitions along a lineage with labels F (foreign), S (Seychelles) such as F-S-F-S will count as two imports and one export, when it is also plausible that it represents a single import with intermediate sequences missing from the sampling. How many similar cases have the authors detected in their analysis? 

I have serious concerns about the validity of the inferences on the number of introductions of the virus based on a single sampling of sequences from GISAID. What is the effect of randomly sampling sequences from abroad on the inference of imports of SARS-CoV-2 into Seychelles? Furthermore, with such a sampling it is not possible to infer the exportation of the virus from the Seychelles to other countries. I think that this should be removed from the manuscript.

The information about the SI in Fig. 1 would be much more valuable if completed with data on vaccination coverage of the population.

This first paragraph in section 3.3 should be moved to 3.4  because it does not describe diversity and its content is directly related to that topic. 

There are several relevant questions related to 3.3 that might be answered by analyzing the data reported. For instance, how much of the genetic variability in the reference population is retained in the Seychelles samples? Is this similar across VOCs/VOIs? What are the main differences between foreign and local sequences of AY.122, the most prevalent lineage in the Seychelles? And so on...

The labels in figure 3 do not correspond with those in the legend.
The figures are too small to observe what the authors describe. It is necessary to provide these images as supplementary information at an appropriate scale or, even better, the corresponding treefiles in Newick format with the relevant labels for identifying the country of origin of each sequence.

Was any of the high frequency mutants in the Seychelles described in section 3.3 very different in the reference sample? Is any of these mutations exclusive or with a likely origin in the Seychelles?

Minor comments

Correct box in Fig. S2: 1298 - 155 = 1143

Author Response

Comments and Suggestions for Authors

Reviewer 1

The manuscript presents an analysis of the genomic epidemiology of SARS-CoV-2 in the Seychelles archipelago based on 1056 complete genome sequences sampled between March 2020 and December 2021. This is very similar work to many other publications on the same topic for different countries, regions, or even cities and as such is an interesting contribution to the global understanding of the introduction and spread of an important emergent virus. However, I think that the work can be improved before it is accepted for publication. My main points for improvement are detailed next.

We thank the reviewer for these comments.

The procedure for estimating the number of virus importations into Seychelles needs to be explained in more detail. According to the current description, a sequence of transitions along a lineage with labels F (foreign), S (Seychelles) such as F-S-F-S will count as two imports and one export, when it is also plausible that it represents a single import with intermediate sequences missing from the sampling. How many similar cases have the authors detected in their analysis? 

We have edited sub section 2.3.7 to explain in detail the procedure for estimating the number of virus importations and exportation. Line 214 - 230

I have serious concerns about the validity of the inferences on the number of introductions of the virus based on a single sampling of sequences from GISAID. What is the effect of randomly sampling sequences from abroad on the inference of imports of SARS-CoV-2 into Seychelles? Furthermore, with such a sampling it is not possible to infer the exportation of the virus from the Seychelles to other countries. I think that this should be removed from the manuscript.

We validated our findings on the number of introductions of the virus, by repeating  the import/export analysis with two sets of data randomly sampled from GISAID. The re-analysis found results that were closely aligned with those revealed by sub-sample one, and thus similar conclusions. We have added text in line 382 – 384 to this effect.

We acknowledge that sampling bias is a potential limitation to our analysis, which is dependent on the completeness of the comparison data from outside Seychelles. This has been captured as a study limitation in line 455.

The information about the SI in Fig. 1 would be much more valuable if completed with data on vaccination coverage of the population.

We have added a statement on vaccination data in line 237-240 and updated Figure 1 to include a plot on vaccination rate in the country.

“The roll out of vaccines in the country appeared to have no effect on the number COVID-19 infections reported in the country i.e., we see a sharp surge of infections in the country due Delta VOC in May-June 2021 period, when % of the population had received first dose of the vaccine”. Studies have reported reduced sensitivity of SARS-CoV-2 Delta VOC to vaccines1. This reduced sensitivity may result in failure to curtail a surge in infections albeit mitigated by preserved protection against severe disease."

This first paragraph in section 3.3 should be moved to 3.4  because it does not describe diversity and its content is directly related to that topic.

We have divided section 3.3 into two sections i.e., section 3.3, which describes the phylogenetic clustering of Seychelles sequences on the world phylogeny, and section 3.4, which describes the genetic diversity and mutations detected in Seychelles sequences

The current section 3.4 of the manuscript has been renamed section 3.5.

There are several relevant questions related to 3.3 that might be answered by analyzing the data reported. For instance, how much of the genetic variability in the reference population is retained in the Seychelles samples? Is this similar across VOCs/VOIs? What are the main differences between foreign and local sequences of AY.122, the most prevalent lineage in the Seychelles? And so on...

On reviewers’ suggestions, we have a carried a number of analyses on our  data set (Line 347 - 352)

  1. We investigated the genetic variation observed in Seychelles lineages in relative to the reference SARS-CoV-2 sequence.
  2. We compared the mutations found in the Seychelles sequences to those found elsewhere in the world.

The labels in figure 3 do not correspond with those in the legend.

This has been corrected.

The figures are too small to observe what the authors describe. It is necessary to provide these images as supplementary information at an appropriate scale or, even better, the corresponding treefiles in Newick format with the relevant labels for identifying the country of origin of each sequence.

The figures have been enlarged and can be included as supplementary materials; however, instead of adding country of origin labels of each the genome used to generate the phylogenetic tree, we have labelled them using continents of origin to avoid over cluttering at the tip labels

Was any of the high frequency mutants in the Seychelles described in section 3.3 very different in the reference sample? Is any of these mutations exclusive or with a likely origin in the Seychelles?

Comparing Seychelles sequences with those from other countries in the world, there was no unique mutations reported only in Seychelles sequences i.e., mutations observed in Seychelles have been observed elsewhere.

Minor comments

Correct box in Fig. S2: 1298 - 155 = 1143

This has been corrected.

Reference

  1. 1. Planas D, Veyer D, Baidaliuk A, et al. Reduced sensitivity of SARS-CoV-2 variant Delta to antibody neutralization. Nature. 2021;596(7871):276-280. doi:10.1038/s41586-021-03777-9

Reviewer 2 Report

In this study, based on the genomic analysis of 1,056 SARS-CoV-2 emerged in Seychelles, authors described the genomic epidemiology of SARS-CoV-2 in Seychelles, and revealed the importance of the introduction of SARS-CoV-2 VOCs for the local surge infection cases in Seychelles. Overall, the conclusions were supported by experimental data, and the manuscript is well written. Nevertheless, some issues should be addressed.

  1. Vaccination coverage and timeliness in Seychelles should be illustrated, which might have great effect on the local surge infection cases;
  2. Line 332, H655Y mutation is not located at S1-S2 furin cleavage site;
  3. Line 330-331, “to reduced sensitivity” is repeated;
  4. Insufficient references to this statement “The Delta VOC, L452R, P681R and D950N mutations, which have been linked to reduced sensitivity to neutralizing antibodies and higher transmissibility (Line 324-326). Actually, the effect of the D950N mutation might not be significant.

Author Response

Comments and Suggestions for Authors

Reviewer 2.

In this study, based on the genomic analysis of 1,056 SARS-CoV-2 emerged in Seychelles, authors described the genomic epidemiology of SARS-CoV-2 in Seychelles, and revealed the importance of the introduction of SARS-CoV-2 VOCs for the local surge infection cases in Seychelles. Overall, the conclusions were supported by experimental data, and the manuscript is well written. Nevertheless, some issues should be addressed.

We thank the reviewer for these comments.

Vaccination coverage and timeliness in Seychelles should be illustrated, which might have great effect on the local surge infection cases;

We have added a statement on vaccination data in line 237-240 and updated Figure 1 to include a plot on vaccination rate in the country.

“The roll out of vaccines in the country appeared to have no effect on the number COVID-19 infections reported in the country i.e., we see a sharp surge of infections in the country due Delta VOC in May-June 2021 period, when 60 % of the population had received first dose of the vaccine”. Studies have reported reduced sensitivity of SARS-CoV-2 Delta VOC to vaccines1. This reduced sensitivity may result in failure to curtail a surge in infections albeit mitigated by preserved protection against severe disease."

Line 332, H655Y mutation is not located at S1-S2 furin cleavage site;

We apologise for this oversight. True, the H665Y is not located on S1-S2 furin cleavage site. We have re-structured the statement  in line 432  as follows

“A cluster of three mutations occurring near the S1-S2 furin cleavage site (H655Y, N679K and P681H) have been associated with increased transmissibility”

Line 330-331, “to reduced sensitivity” is repeated;

Repeated text has been removed.

Insufficient references to this statement “The Delta VOC, L452R, P681R and D950N mutations, which have been linked to reduced sensitivity to neutralizing antibodies and higher transmissibility (Line 324-326). Actually, the effect of the D950N mutation might not be significant.

We had added 1 more reference to this statement.

Study by Planas et al 20211 which has shown reduced sensitivity of SARS-CoV-2 Delta VOC to neutralizing antibodies, The study demonstrates roles played by L452R and P681R mutations on reduced sensitivity to neutralizing antibodies elicited by previous infection with SARS-CoV-2 or by vaccination.

We have deleted D950N mutation from this statement (line 425) since it has not been extensively characterised, and little is known about its role in the spike gene.

Reference

  1. 1. Planas D, Veyer D, Baidaliuk A, et al. Reduced sensitivity of SARS-CoV-2 variant Delta to antibody neutralization. Nature. 2021;596(7871):276-280. doi:10.1038/s41586-021-03777-9

Round 2

Reviewer 1 Report

The authors have adressed appropiately my concerns and have completed some additional analyses that were demanded.

Reviewer 2 Report

In the revised version, authors have addressed my question.